# Retrieving Signals in the Frequency Domain with Deep Complex Extractors

## Abstract

Recent advances have made it possible to create deep complex-valued neural networks. Despite this progress, the potential power of fully complex intermediate computations and representations has not yet been explored for many challenging learning problems. Building on recent advances, we propose a novel mechanism for extracting signals in the frequency domain. As a case study, we perform audio source separation in the Fourier domain. Our extraction mechanism could be regarded as a local ensembling method that combines a complex-valued convolutional version of Feature-Wise Linear Modulation (FiLM) and a signal averaging operation. We also introduce a new explicit amplitude and phase-aware loss, which is scale and time invariant, taking into account the complex-valued components of the spectrogram. Using the Wall Street Journal Dataset, we compare our phase-aware loss to several others that operate both in the time and frequency domains and demonstrate the effectiveness of our proposed signal extraction method and proposed loss. When operating in the complex-valued frequency domain, our deep complex-valued network substantially outperforms its real-valued counterparts even with half the depth and a third of the parameters. Our proposed mechanism improves significantly deep complex-valued networks' performance and we demonstrate the usefulness of its regularizing effect.

## 1 Introduction

Complex-valued neural networks have been studied since long before the emergence of modern deep learning techniques (Georgiou & Koutsougeras, 1992; Zemel et al., 1995; Kim & Adalı, 2003; Hirose, 2003; Nitta, 2004). Nevertheless, deep complex-valued models have only started to gain momentum (Reichert & Serre, 2014; Arjovsky et al., 2015; Danihelka et al., 2016; Trabelsi et al., 2017; Jose et al., 2017; Wolter & Yao, 2018b; Choi et al., 2019), with the great majority of models in deep learning still relying on real-valued representations. The motivation for using complex-valued representations for deep learning is twofold: On the one hand, biological nervous systems actively make use of synchronization effects to gate signals between neurons – a mechanism that can be recreated in artificial systems by taking into account phase differences (Reichert & Serre, 2014). On the other hand, complex-valued representations are better suited to certain types of data, particularly those that are naturally expressed in the frequency domain.

Other benefits provided by working with complex-valued inputs in the spectral or frequency domain are computational. In particular, short-time Fourier transforms (STFTs) can be used to considerably reduce the temporal dimension of the representation for an underlying signal. This is a critical advantage, as training recurrent neural networks (RNNs) or convolutional neural networks (CNNs) on long sequences remains challenging due to unstable gradients and the computational requirements of backpropagation through time (BPTT) (Hochreiter, 1991; Bengio et al., 1994). Applying the STFT on the raw signal, on the other hand, is computationally efficient, as in practice it is implemented with the fast Fourier transform (FFT) whose computational complexity is $\mathcal{O}(n\,log(n))$.

The aforementioned biological, representational and computational considerations provide compelling motivations for designing learning models for tasks where the complex-valued representation of the input and output data is more desirable than their real-counterpart. Recent work has provided building blocks for deep complex-valued neural networks (Trabelsi et al., 2017). These building blocks have been shown, in many cases, to avoid numerical problems during training and, thereby, enable the

use of complex-valued representations. These representations are well-suited for frequency domain signals, as they have the ability to explicitly encode frequency magnitude and phase components. This motivates us to design a new signal extraction mechanism operating in the frequency domain. In this work, our contributions are summarized as follows:

1. We present a new signal separation mechanism implementing a local ensembling procedure. More precisely, a complex-valued convolutional version of Feature-wise Linear Modulation (FiLM) (Perez et al., 2018) is used to create multiple separated candidates for each of the signals we aim to retrieve from a mixture of inputs. A signal averaging operation on the candidates is then performed in order to increase the robustness of the signal to noise and interference. Before the averaging procedure, a form of dropout is implemented on the signal candidates in order to reduce the amount of interference and noise correlation existing between the different candidates.

2. We propose and explore a new magnitude and phase-aware loss taking **explicitly** into account the magnitude and phase of signals. A key characteristic of our loss is that it is scale- and time-invariant.

We test our proposed signal extraction mechanism in the audio source separation setting where we aim to retrieve distinct audio signals associated with each speaker in the input mix. Our experiments demonstrate the usefulness of our extraction method, and show its **regularizing effect**.

## 2 RELATED WORK

### 2.1 RELATED WORK ON LEARNING REPRESENTATIONS IN THE FOURIER DOMAIN

Leveraging the Convolution Theorem to retrieve information has been done decades ago in the machine learning community using holographic reduced representations (HRRs) in the context of associative memories (Plate, 1991; 1995). HRRs enable one to store key-value data. Retrieval of a value in the data associated with a given key can be performed by convolving the whole data with the key or by applying an inner product between these two. By applying a fast Fourier transform (FFT) on the keys and the data, one can perform elementwise multiplication between the Fourier transforms and apply an inverse FFT to convert the result to the time domain. This would be equivalent to performing circular convolution between the key and the data in the time domain and has the advantage of being less expensive. Recently, Danihelka et al. (2016) have used associative memories to augment the capacity of LSTMs and to increase their robustness to noise and interference. For that, they applied independent permutations on the memory to create multiple copies of it. This enables one to obtain decorrelated noise in each of the permuted copies. A complex multiplication is then performed between the key and each of the copies. A signal averaging on the resulted multiplications eliminates the decorrelated noise in them and strengthens the signal-to-noise ratio (SNR) of the retrieved signal. Danihelka et al. (2016), however, have not relied on FFTs in order to convert the temporal signals to the frequency domain. In fact, they assumed that complex-valued multiplication between the key and the data is itself enough to perform retrieval, and they have assumed that for each input representation the first half is real and the second one is imaginary.

During this decade, interest in Fourier domain representations has started to grow in the machine learning community. Bruna et al. (2013) introduced a generalization of convolutions to graphs using the Graph Fourier Transform, which is defined as the multiplication of a graph signal by the eigenvector matrix of the graph Laplacian. However, the computation of the eigenvector matrix is expensive. Recently, methods that are computationally more efficient have been introduced in Defferrard et al. (2016) and Kipf & Welling (2016) to avoid an explicit use of the Graph Fourier basis. In the context of Convolutional Neural Networks (CNNs), Rippel et al. (2015) introduced spectral pooling, which allows one to perform pooling in the frequency domain. This allows one to maintain the output spatial dimensionality, and thus the technique can retain significantly more information than other pooling approaches. Rippel et al. (2015) have also observed that the parametrization of the convolution filters in the Fourier domain induces faster convergence during training. Arjovsky et al. (2016) designed a recurrent neural network (RNN) where the transition hidden matrix is unitary. More precisely, the hidden transition matrix is constructed using the product of specific unitary transformations such as diagonal matrices, permutations, rotations, the Discrete Fourier Transform and its inverse. This allows one to preserve the norm of the hidden state, and as a consequence,

mitigates the problem of vanishing and exploding gradients. Wolter & Yao (2018a) designed an RNN where the input is converted to the frequency domain using a Short Time Fourier Transform (STFT). The output is converted back to the time domain by applying an inverse STFT. Zhang et al. (2018) proposed a Fourier Recurrent Unit (FRU) where they showed that FRU has gradient lower and upper bounds independent of the temporal dimension. They have also demonstrated the great expressivity of the sparse Fourier basis from which the FRU draws its power. As we consider the task of speech separation as case study, we provide a related work section on both time domain and frequency domain speech separation methods in section 2.2.

## 2.2    RELATED WORK ON TIME DOMAIN AND FREQUENCY DOMAIN SPEECH SEPARATION

Speech separation has been the subject of extensive study within the audio processing literature for a considerable amount of time. Recently, there has been growing interest in leveraging deep learning techniques (Du et al., 2014; Huang et al., 2014; Hershey et al., 2015; Gao et al., 2018; Ephrat et al., 2018) to tackle the speech separation problem. Hershey et al. (2015) proposed a deep clustering approach to speech separation. The basic idea is to learn high-dimensional embeddings of the mixture signals, that is later exploited to separate the speech targets with standard clustering techniques. A recent attempt to extend deep clustering led to the deep attractor network proposed by Chen et al. (2016). Similarly to deep clustering, high dimensional embeddings are learned, but the network also creates the so-called "attractors" to better cluster time-frequency points dominated by different speakers. The aforementioned approaches *estimate only the magnitude of the STFTs* and reconstruct the time-domain signal with the Griffin-Lim algorithm (Griffin & Lim, 1984) or other similar procedures (Sturmel & Daudet, 2006). Other papers have recently proposed to integrate the phase-information within a speech separation system without necessarily working in the complex-valued frequency domain. The work by Erdogan et al. (2015), for instance, proposes to train a deep neural network with a phase-sensitive loss. Another noteworthy attempt has been described in Wang et al. (2018), where the neural network still estimates the magnitude of the spectrum, but the time-domain speech signals are retrieved by directly integrating the Griffin-Lim reconstruction into the neural layers. Furthermore, methods reported in Wang et al. (2018) integrate the phase-information within a speech separation system by reconstructing the clean phase of each source starting from the estimated magnitude of each source and the mixture phase. This is fundamentally different from our proposed framework, as we provide an end-to-end solution to perform signal retrieval in the complex-valued frequency domain, and process both spectrogram magnitude and phase information rather than working only on magnitude representation with heuristic reconstruction of phase. Another attempt to estimate the clean phase is reported in Le Roux et al. (2019) where the clean phase of each speaker is estimated using discrete representation. This is also fundamentally different from our work as it considers a discrete representation of the phase for source separation and, in our case, we consider continuous representation of the complex-domain signal.

Instead of explicitly integrating phase-information, other recent work perform speech separation *in the time domain directly*, as described in Venkataramani & Smaragdis (2018). Likewise, the TasNet architecture proposed in Luo & Mesgarani (2017) and ConvTasNet (Luo & Mesgarani, 2018) perform speech separation using the mixed time signal as input. Operating directly on the time-domain signal using the ConvTasNet architecture, which implements temporal convolutional networks (TCN) (Bai et al., 2018), has led to state-of-the-art results in audio speech separation (Luo & Mesgarani, 2018; Shi et al., 2019). The studies by (Lee et al., 2017; Hu & Wang, 2004; Huang et al., 2014) are more related to our work as they address the speech separation problem by processing the complex-valued spectral input of the mixed speech. However, this was done without leveraging the recent advances in complex-valued deep learning.

## 3    CONNECTION TO SIGNAL PROCESSING: MOTIVATION FOR USING FiLM AND SIGNAL AVERAGING

Our signal extraction method takes advantage of the convolution theorem which states that the Fourier transform of two circularly convolved signals is the elementwise product of their Fourier transforms. It also implements a signal averaging procedure that allows to reduce the energy of the noise existing in the estimates of a clean signal, and so, to increase their respective signal to noise ratio (SNR). We detail here the motivation for using FiLM (Perez et al., 2018) and how our extraction method

increases the signal to noise ratio. Let's consider a clean signal $\mathbf{s}$ corrupted by the environment impulse response $\mathbf{r}$ and an additive noise $\epsilon$. The corrupted signal can be expressed as $\mathbf{y} = \mathbf{s} \circledast \mathbf{r} + \epsilon$, where $\circledast$ denotes the circular convolution operator. By leveraging the convolution theorem and the linearity of the Fourier transform we get :

$$\mathcal{F}(\mathbf{y}) = \mathcal{F}(\mathbf{s}) \odot \mathcal{F}(\mathbf{r}) + \mathcal{F}(\epsilon), \tag{1}$$

where $\mathcal{F}$ denotes the Fourier transform and $\odot$ the complex element-wise multiplication. If we want to retrieve the spectral information of the clean signal $\mathbf{s}$, we can express it as:

$$\mathcal{F}(\mathbf{s}) = \left[ \mathcal{F}(\mathbf{y}) \odot \frac{1}{\mathcal{F}(\mathbf{r})} \right] - \frac{\mathcal{F}(\epsilon)}{\mathcal{F}(\mathbf{r})}, \tag{2}$$

where $\frac{1}{\mathcal{F}(\mathbf{r})}$ and $-\frac{\mathcal{F}(\epsilon)}{\mathcal{F}(\mathbf{r})}$ are respectively scaling and shifting representations. These representations could easily be inferred using FiLM (Perez et al., 2018) as it conditionally learns scaling $\mathbf{\Gamma}$ and shifting $\mathbf{B}$ representations. To be more rigorous, we can assume in the case of speech separation that, for each speaker, there exists an impulse response such that when it is convolved with the clean speech of the speaker, it allows to reconstruct the mix. We would then have:

$$\mathbf{mix} = \mathbf{s}_i \circledast \mathbf{r}_i + \epsilon_i \quad \forall i \in \{1, ..., \text{Nb speakers}\}$$
$$\Rightarrow \mathcal{F}(\mathbf{s}_i) = \mathcal{F}(\mathbf{mix}) \odot \frac{1}{\mathcal{F}(\mathbf{r}_i)} - \frac{\mathcal{F}(\epsilon_i)}{\mathcal{F}(\mathbf{r}_i)} \tag{3}$$
$$\Rightarrow \mathcal{F}(\mathbf{s}_i) = \mathcal{F}(\mathbf{mix}) \odot \mathbf{\Gamma}_i + \mathbf{B}_i.$$

Now, let's assume that $\mathbf{y}$ is a stochastic process such that $\mathbf{y} = \mathbf{x} + \epsilon$, where $\epsilon$ is the noise component which mean $\mathbb{E}[\epsilon] = 0$. $\mathbf{x}$ is the clean signal that we want to estimate such that $\mathbf{x}$ is constant for all observations and that an $i^{th}$ observation of $\mathbf{y}$ is given by $\mathbf{y}_i = \mathbf{x} + \epsilon_i$. The signal-to-noise ratio (SNR), which is a measure of the signal quality, is defined as the ratio of the power of a clean signal to the power of noise, i.e, $\text{SNR} = \frac{\mathbb{E}[|\mathbf{x}|^2]}{\mathbb{E}[|\epsilon|^2]}$. Estimating the clean speech $\mathbf{x}$ by approximating $\mathbb{E}[\mathbf{y}]$ allows to discard the noise component as $\mathbb{E}[\mathbf{y}] = \mathbf{x}$. In that case $\hat{\mathbf{x}} = \frac{1}{N} \sum_{i=1}^{N} (\mathbf{x} + \epsilon_i) = \mathbf{x} + \frac{1}{N} \sum_{i=1}^{N} \epsilon_i$. The SNR would then be: $\text{SNR} = \frac{\mathbb{E}[|\mathbf{x}|^2]}{\mathbb{E}[|\frac{1}{N} \sum_{i=1}^{N} \epsilon_i|^2]} = \frac{\mathbb{E}[|\mathbf{x}|^2]}{\frac{1}{N^2} \mathbb{E}[|\sum_{i=1}^{N} \epsilon_i|^2]}$. If $\epsilon_i$ are uncorrelated, $\mathbb{E}[|\sum_{i=1}^{N} \epsilon_i|^2] = \sum_{i=1}^{N} \mathbb{E}[|\epsilon_i|^2] = N \mathbb{E}[|\epsilon_i|^2] \Rightarrow \text{SNR} = N \frac{\mathbb{E}[|\mathbf{x}|^2]}{\mathbb{E}[|\epsilon_i|^2]}$. This shows that the signal averaging operation and the uncorrelated noises allow to increase the SNR by a factor of $N$. If we want to approximate $\mathcal{F}(\mathbf{s}_i)$ by performing signal averaging, we would then have:

$$\mathbb{E}[\mathcal{F}(\mathbf{s}_i)] = \mathcal{F}(\mathbf{mix}) \odot \mathbb{E}[\mathbf{\Gamma}_i] + \mathbb{E}[\mathbf{B}_i]$$
$$\Rightarrow \widehat{\mathbb{E}[\mathcal{F}(\mathbf{s}_i)]} = \mathcal{F}(\mathbf{mix}) \odot \widehat{\mathbb{E}[\mathbf{\Gamma}_i]} + \widehat{\mathbb{E}[\mathbf{B}_i]} \tag{4}$$
$$= \mathcal{F}(\mathbf{mix}) \odot \frac{1}{N} \sum_{j=1}^{N} \mathbf{\Gamma}_{ij} + \frac{1}{N} \sum_{j=1}^{N} \mathbf{B}_{ij},$$

where $\mathcal{F}(\mathbf{mix})$ is constant. In equation (4), $N$ is equal to the number of scaling and shifting representations generated to approximate respectively each of $\mathbb{E}[\mathbf{\Gamma}_i]$ and $\mathbb{E}[\mathbf{B}_i]$.

## 4   AMPLITUDE AND PHASE-AWARE LOSS

In Choi et al. (2019) a weighted version of the cosine similarity is proposed in order to maximize the signal-to-distortion ratio (SDR) proposed in Vincent et al. (2006). Recall that cosine similarity loss is defined in the real-valued domain and it is given by the following equation:

$$\cos_{\text{time}}(\mathbf{y}, \mathbf{x}) = \frac{- \sum_i x_i \circ y_i}{||\mathbf{x}|| \cdot ||\mathbf{y}||}, \tag{5}$$

where $\circ$ denotes the element-wise real-valued multiplication operation. Both $\mathbf{y}$ and $\mathbf{x}$ are real-valued in the above equation as $\mathbf{y}$ is the target signal in the temporal domain and $\mathbf{x}$ is the estimated signal after performing an inverse STFT on the spectrogram. The phase is then taken **implicitly** into account as the real-valued target signal encodes inherently the phase of the spectrogram. As the task in Choi et al. (2019) is speech enhancement (*which is different from ours as we are performing speech separation*), the authors used a weighted version of the $\cos_{\text{time}}$ loss to weight the part of the loss

corresponding to the speech signal and also the complementary part corresponding to the noise signal. This weighting is performed according to their respective target energies. In our case we are interested in extracting the clean speech signals of all the involved speakers whether each speaker signal has either high or low energy in the mixture. This is why we are not interested in penalizing the retrieved speech of each speaker by its corresponding energy.

Here, we suggest the use of a loss function which explicitly takes into account both magnitude and phase. This is accomplished by computing the inner product, between the reference signal and its estimate, in the complex plane. In fact computing the inner product in the frequency domain is equivalent to computing the cross correlation in the time domain followed by a weighted average. The inner product in the frequncy domain is then shift-invariant (time-invariant). The complex inner product between 2 signals is given by the following equation:

$$\langle \mathbf{x} | \mathbf{y} \rangle = \sum_j [\Re(x_j)\Re(y_j) + \Im(x_j)\Im(y_j)] + i \left[\Re(x_j)\Im(y_j) - \Im(x_j)\Re(y_j)\right]. \tag{6}$$

If $\mathbf{x}$ and $\mathbf{y}$ are identical, which is equivalent of having $||\mathbf{x}|| = ||\mathbf{y}||$ and $\angle \mathbf{x} = \angle \mathbf{y}$, then, $\langle \mathbf{x} | \mathbf{y} \rangle = ||\mathbf{y}||^2 + 0i$. If $\mathbf{x}$ and $\mathbf{y}$ are parallel, then $\frac{\langle \mathbf{x} | \mathbf{y} \rangle}{||\mathbf{x}|| \cdot ||\mathbf{y}||} = 1 + 0i = 1$. The inner product between the 2 signals normalized by the product of their amplitudes, is then scale and time invariant. We chose a loss that maximizes the real part of that normalized inner product and minimizes the square of its imaginary part. Note that each of the real and imaginary parts of the normalized inner product lies between [-1, 1]. We refer the reader to section A.1 in the appendix for more information on how the complex inner product is both amplitude and phase aware, how the real part of equation (6) is responsible of the amplitude similarity between the reference and estimate signals and how the imaginary part of the same equation is responsible for the phase matching between them. We define the following similarity loss denoted by $\mathbb{C}\text{SimLoss}$ as:

$$\mathbb{C}\text{SimLoss}(\mathbf{x}, \mathbf{y}) = -\lambda_{real} \Re \left( \frac{\langle \mathbf{x} | \mathbf{y} \rangle}{||\mathbf{x}|| \cdot ||\mathbf{y}||} \right) + \lambda_{imag} \Im^2 \left( \frac{\langle \mathbf{x} | \mathbf{y} \rangle}{||\mathbf{x}|| \cdot ||\mathbf{y}||} \right), \tag{7}$$

where $\lambda_{real}$ and $\lambda_{imag}$ are penalty constants. $\lambda_{real}$ is penalizing amplitude mismatch and $\lambda_{imag}$ is penalizing phase mismatch. We fixed $\lambda_{real}$ to 1 in all our experiments. We tried different values of $\lambda_{imag} \in \{10^2, 10^3, 10^4\}$. we found that the only value of $\lambda_{imag}$ that allows the phase matching part of the train loss to have same range of values than the amplitude matching part is $\lambda_{imag} = 10^4$. All the results are reported in Table 2 and Table 3 for $\mathbb{C}\text{SimLoss}$ correpond to $\lambda_{imag} = 10^4$.

## 5 DETAILS OF THE U-NET ARCHITECTURE USED FOR SPEECH SEPARATION

We detail here the architecture we used to perform speech separation[1]. For this, we rely on the U-Net architecture proposed by Ronneberger et al. (2015) and the complex-valued building blocks proposed by Trabelsi et al. (2017). This is similar to the complex-valued U-Net architecture used in Dedmari et al. (2018) who reported state-of-the-art results in MRI reconstruction using complex-valued raw input. Our primary goal is to demonstrate that our proposed signal extraction mechanism can improve upon the performance of baseline models.

For our task, we required the addition of residual connections inside the U-Net blocks and replaced complex batch normalization with complex layer normalization, as the model was otherwise unable to learn, yielding instabilities during training. The reasons why complex LayerNorm outperformed complex BatchNorm are discussed in the appendix in section A.2. We describe, in section 6 how our extraction mechanism is implemented in the context of audio source separation.

### 5.1 COMPLEX RESIDUAL U-NET

Residual networks (He et al., 2016a) and identity connections (He et al., 2016b) have had a significant impact on image segmentation. These architectural elements have also been combined with U-Nets (Drozdzal et al., 2016) for image segmentation. In our case, we use simple basic complex residual blocks (Figure 2 in appendix) inside each of the U-Net encoding and decoding paths (Figure 1) . Figure 2 (Left) and (Middle) illustrate the basic structure of our Residual U-Net upsampling and

---

[1]The source code is located at https://github.com/FourierSignalRetrievalICLR2020/FourierExtraction

downsampling blocks ($U_i$ and $D_i$) used in Figure 1, while Figure 2 (Right) illustrates the structure of the complex residual blocks used in Figure 2 (Left) and Figure 2 (Middle).

Each U-Net block begins with a downsampling block (in the encoding U-Net path) or an upsampling block (in the decoding U-Net path). It also contains a block that doubles the number of feature maps (in the encoding path), or halves them (in the decoding path). The upsampling, downsampling, doubling and halving blocks each applies successively a complex layer normalization, a $\mathbb{C}$ReLU and a complex convolution to their inputs. All complex convolutions have a kernel size of $3 \times 3$ except for the case of a downsampling block, where the convolution layer has a kernel size of $1 \times 1$ and a stride of $2 \times 2$. In the case of upsampling, we use bilinear interpolation instead of transposed convolution because we found empirically that it yielded better results. Immediately before and immediately after the doubling / halving blocks, we use $k = 1$ or $k = 2$ residual blocks. We have opted for this residual U-Net block architecture because of memory constraints and because residual connections are believed to perform inference through iterative refinement of representations (Greff et al., 2016; Jastrzebski et al., 2017).

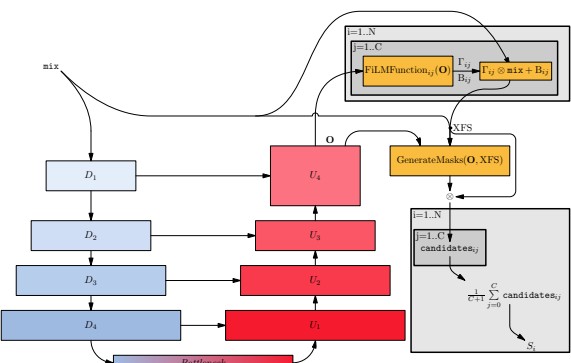

Figure 1: **The architecture of our Deep Complex Extractor**. It consists of a pipeline containing a U-Net and a Complex Extractor Masking operator (see Algorithm 1). The Deep Extractor takes as input the mixed speech signal which is fed to the U-Net. The downsampling blocks of the U-Net are denoted by $D_i$ where $i \in \{1, 2, 3, 4\}$ and the upsampling blocks are denoted by $U_i$ where $i \in \{1, 2, 3, 4\}$. The output of the U-Net along with the input mix are then fed to the Complex Extractor Masking operator in order to estimate the clean speech for each of the speakers.

## 6 COMPLEX MASK GENERATION

Featurewise Linear Modulation (FiLM) (Perez et al., 2018) techniques have yielded impressive results in visual question answering (VQA). The FiLM approach applies an affine transformation to convolutional feature maps, given the embedding of the question. In our approach, we create multiple transformations of the complex input spectrogram using FiLM. The FiLM parameters are determined from the output of our U-Net (**See Figure 1**). We then generate a complex mask for the original input spectrogram as well as for each of the FiLM-transformed spectrograms. This is accomplished by using a ResNet conditioned on the U-Net output, the spectrogram and its FiLM transformations. Each spectrogram is multiplied by its corresponding complex mask. This leads to multiple candidates for the separated speech of each speaker. The resulting outputs are averaged to produce the final estimated clean speech. This could be interpreted as a local ensembling procedure to estimate the clean speech of the different speakers. More precisely, given the output of the last upsampling block of the U-Net, we generate scaling matrices $\Gamma_j$ and shift matrices $B_j$, $j \in [1, C]$ of the same size as the input mix spectrogram. These parameters operate on the input mix as described by the following equation:

$$\text{input\_transformation}_j = \Gamma_j \otimes \text{inputmix} + B_j, \tag{8}$$

where $\Gamma_j$ and $B_j$ are functions of the output of the last upsampling block in the U-Net, and $\otimes$ is the elementwise complex product. In our case, we used a simple complex convolution layer with a kernel of size $3 \times 3$ to generate $\Gamma_j$ and $B_j$. The original input mix and its $C$ scaled and shifted

transformations together form $C + 1$ representations of the input mix. Given these $C + 1$ complex representations, we generate $C + 1$ corresponding complex masks, with which the representations are then multiplied. These masks are generated by a sequence of a complex convolution layer which kernel size is $3 \times 3$ followed by two residual blocks. Once we have performed the complex multiplication of the masks with their respective inputs, $C + 1$ separated speech candidates are obtained for a given speaker. This procedure is repeated for the maximum number of speakers that could exist in an input mix. The main motivation for this process is to increase the separation capability and reduce interference between the separated speakers. Each transformation can focus on a specific pattern in the representation. Each mask corresponding to a specific input transformation can be seen as a feature of the speaker embedding. Grouped together, the masks generated to retrieve the speech of a given speaker could be interpreted as an embedding identifying the speaker. The proposed complex masking procedure is summarized in **Algorithm 1**.

---

**Algorithm 1 Complex Extractor Masking**

---

**Input:** *U-Net output:*      O
**Input:** *Nb transformations (XFs):*   C
**Input:** *Nb speakers:*      N
**Input:** *Input Mix:*      mix
**Output:** *Speakers separated speeches:* $S_1, ..., S_N$

1: **function** $\mathbb{C}$-FILMED MASKING(O, $C$, $N$, mix)
2:      **for** $i \leftarrow 1$ **to** $N$ **do**
3:          $\Gamma_{i1} ... \Gamma_{iC}$, $\mathrm{B}_{i1} ... \mathrm{B}_{iC} \leftarrow$ FilMFunction(O)
4:      **end for**
5:      XFS $\leftarrow$ [ ]
6:      **for** $i \leftarrow 1$ **to** $N$ **do**
7:          **for** $j \leftarrow 1$ **to** $C$ **do**
8:              $\mathrm{XF}_{ij} \leftarrow \Gamma_{ij} \otimes \mathrm{mix} + \mathrm{B}_{ij}$
9:              $\mathrm{XFS}_i$.append($\mathrm{XF}_{ij}$)
10:          **end for**
11:      **end for**
12:      XFS $\leftarrow$ concatenate($\mathrm{XFS}_{11}, ..., \mathrm{XFS}_{NC}$)
13:      masks $\leftarrow$ GenerateMasks(O, XFS)
14:      candidates $\leftarrow$ masks $\otimes$ XFS
15:      cleanspeeches $\leftarrow$ [ ]
16:      **for** $i \leftarrow 1$ **to** $N$ **do**
17:          $\mathrm{cleanspeech}_i \leftarrow$ average(candidates$[(C + 1) \times (i - 1) + 1 : (C + 1) \times i])$)
18:          cleanspeeches.append($\mathrm{cleanspeech}_i$)
19:      **end for**
20:      **return** cleanspeeches
21: **end function**

---

## 7   SYNOPSIS OF THE EXPERIMENTS

We present in Table 1 the most important results obtained when conducting our experiments. The complete results and the extended empirical analysis can be found in the appendix in section A.5. The data pre-processing and training protocol can be found in the appendix, in section A.4.

We explore several variants of our architecture and report the test SDR. They are parametrized by:

- $k$, the number of residual blocks used inside the residual U-Net block (See Figure 2).
- START FMAPS, the number of feature maps in the first layer of the encoding path in the U-Net. START FMAPS defines the depth of each of the successive layers in the model. [2]
- PARAMS, the number of parameters, in millions.
- TRANSFORMS, the number of input mixture transformations.
- DROPOUT, the mask dropout rate. [3]

---

[2] The effective number of feature maps for a complex feature map is equal to the number of reported feature maps $\times$ 2. This is due to the fact that it has a real and an imaginary part.

[3] Dropping out a mask is equivalent to a dropout of input mixture transformations or clean speech candidates. Performing dropout on the masks reduces the correlation of the different noise components existing in the different candidates of clean speech. Along with signal averaging, dropout regularizes the retrieval mechanism.

Table 1: The most important results obtained for the task of speech separation conducted on mixtures of 2 speakers using the standard setup with the Wall Street Journal corpus.

| MODEL | $k$ | START FMAPS | PARAMS | TRANSFORMS | DROPOUT | LOSS | TEST SDR |
|---|---|---|---|---|---|---|---|
| REAL U-NET | 1 | 64 | 8.45 | 0 | 0 | $L2_{freq}$ | 4.59 |
| REAL U-NET | 2 | 64 | 14.76 | 0 | 0 | $L2_{freq}$ | 7.92 |
| COMPLEX U-NET | 1 | 32 | 4.29 | 0 | 0 | $L2_{freq}$ | 9.61 |
| COMPLEX U-NET | 2 | 32 | 7.4 | 0 | 0 | $L2_{freq}$ | **9.70** |
| COMPLEX U-NET | 2 | 40 | 11.67 | 15 | 0 | $L2_{freq}$ | **10.93** |
| COMPLEX U-NET | 2 | 40 | 11.67 | 15 | 0 | $\mathbb{C}$SimLoss | 10.91 |
| COMPLEX U-NET | 2 | 40 | 11.61 | 10 | 0 | $L2_{time}$ | 10.86 |
| COMPLEX U-NET | 2 | 40 | 11.67 | 15 | 0 | $\cos_{time}$ | 10.74 |
| COMPLEX U-NET | 2 | 44 | 13.97 | 0 | 0 | $L2_{freq}$ | 9.88 |
| COMPLEX U-NET | 2 | 44 | 13.97 | 0 | 0 | $\mathbb{C}$SimLoss | 9.87 |
| COMPLEX U-NET | 2 | 44 | 14.09 | 15 | 0.1 | $L2_{freq}$ | 10.91 |
| COMPLEX U-NET | 2 | 44 | 14.03 | 10 | 0.1 | $\mathbb{C}$SimLoss | 11.34 |

From the first four rows of the results contained in Table 1, we will highlight that the complex-valued baseline models vastly outperform their real-valued counterparts. These baselines (both real and complex) are architecturally the same as the U-Net of Figure 1, but do not include our extraction mechanism (the FiLM, GenerateMask and signal-averaging operations). The real and complex U-Nets' outputs are masks that are complex-multiplied with the mix to infer the clean speech of the speakers. All complex models, whether they have approximately the same number of parameters ($\mathbb{R}$:8.45M $\approx$ $\mathbb{C}$:7.4M), half ($\mathbb{R}$:8.45M; $\mathbb{C}$:4.39M) or a third, with half the depth ($\mathbb{R}$:14.76M; $\mathbb{C}$:4.39M) outperformed by a convincing margin their real counterparts. Thus, natively-complex input, inference and output give complex networks such an overwhelming advantage that almost no handicap of size or depth can mask it. We will therefore not consider real-valued models, transformations and losses any further.

A second highlight from Table 1 is that, while our signal extraction mechanism is inexpensive in terms of parameter count, the extraction mechanism substantially improves the quality of the retrieved signal. For instance, when 10 mixture transformations are in use, the number of parameters is marginally increased by less than 1% (13.97M to 14.03M) while a substantial jump in terms of SDR is observed (from 9.87 to 11.34). This can be also observed in Figure 3 in the appendix. Dropping out the speech candidates with low probability has a further regularization effect on the wider models that have more feature maps, as shown in appendix Figure 5.

The third highlight is that spectral-domain losses, i.e $\mathbb{C}$SimLoss and $L2_{freq}$, outperform their time-domain counterparts. Our proposed $\mathbb{C}$SimLoss posts the best reported result, 11.34 SDR, and Tables 2 and 3 and Figure 4 demonstrate that our extraction mechanism is ideally paired with the $\mathbb{C}$SimLoss objective.

Finally, and although this is out of scope for this paper, we compare ourselves to ConvTasNet (Luo & Mesgarani, 2018), which operates on a time-domain input mixture (as mentioned in §2.2). ConvTasNet claims state-of-the-art results in speech separation, and has led to even further improvements (Shi et al., 2019). Its headline achievement of 15.6 SDR must, however, be understood in light of a significant difference in their preparation of the dataset. Whereas we follow the standard setup described in Hershey et al. (2015), with input mixtures generated using an SNR between 0 and 5 dB, Luo & Mesgarani (2018) use an SNR between -5 and 5 dB. Keeping this in mind, we retrain optimally-configured ConvTasNet but using the standard setup, and obtain 12.1 SDR, compared to our own model's 11.3 SDR.

## 8   CONCLUSION

In this work, we introduced a new complex-valued extraction mechanism for signal retrieval in the Fourier domain. As a case study, we considered audio source separation. We also proposed a new phase-aware loss taking, explicitly, into account the magnitude and phase of the reference and estimated signals. The amplitude and phase-aware loss improves over other frequency and time-domain losses. We believe that our proposed method could lead to new research directions where signal retrieval is needed.

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

## A  APPENDIX

### A.1  DETAILS ABOUT THE AMPLITUDE AND PHASE-AWARE LOSS

We show here that solving the two-equation system, assimilating the real part of the inner product, between the two signals $\mathbf{x}$ and $\mathbf{y}$, to the square of the amplitude of $\mathbf{y}$, and canceling its imaginary part, amounts to canceling the differences in amplitude and phase between $\mathbf{x}$ and $\mathbf{y}$, respectively (See equation 11). For this we will use the following trigonometric properties:

$$\begin{cases} \cos(\theta_x)\cos(\theta_y) &= \frac{1}{2}\cos(\theta_x - \theta_y) + \frac{1}{2}\cos(\theta_x + \theta_y) \\ \sin(\theta_x)\sin(\theta_y) &= \frac{1}{2}\cos(\theta_x - \theta_y) - \frac{1}{2}\cos(\theta_x + \theta_y) \\ \cos(\theta_x)\sin(\theta_y) &= \frac{1}{2}\sin(\theta_x + \theta_y) - \frac{1}{2}\sin(\theta_x - \theta_y) \\ \sin(\theta_x)\cos(\theta_y) &= \frac{1}{2}\sin(\theta_x + \theta_y) + \frac{1}{2}\sin(\theta_x - \theta_y), \end{cases} \tag{9}$$

where $\theta_x, \theta_y \in \mathbb{R}$. For simplicity of notation, we will denote a complex-valued target scalar as $y$ and a its estimate as $x$ instead of $\hat{y}$:

$$
\begin{aligned}
y &= |y|e^{i\theta_y} = |y|\left[\cos(\theta_y) + i\sin(\theta_y)\right] \in \mathbb{C} \\
x &= |y|e^{i\theta_x} = |x|\left[\cos(\theta_x) + i\sin(\theta_x)\right] \in \mathbb{C}.
\end{aligned} \tag{10}
$$

$\theta_y$ and $\theta_x$ are the corresponding phases of the reference $y$ and its complex estimate $x$ respectively. Resolving the system of equations below is equivalent of having both magnitude and phase of the reference and estimation identical **OR** when $y$ is 0. Recall that $\Re(\langle \mathbf{x}|\mathbf{y}\rangle) = \sum_j [\Re(x_j)\Re(y_j) + \Im(x_j)\Im(y_j)]$ and $\Im(\langle \mathbf{x}|\mathbf{y}\rangle) = \sum_j [\Re(x_j)\Im(y_j) - \Re(y_j)\Im(x_j)]$.

$$
\begin{cases}
\Re(x)\Re(y) + \Im(x)\Im(y) - |y|^2 = 0 \\
\Re(x)\Im(y) - \Re(y)\Im(x) = 0
\end{cases}
$$

$$
\Leftrightarrow \begin{cases}
\Re(x)\Re(y) + \Im(x)\Im(y) = \Re(y)^2 + \Im(y)^2 \\
\Re(x)\Im(y) = \Re(y)\Im(x)
\end{cases}
$$

$$
\Leftrightarrow \begin{cases}
|x|\cos(\theta_x)|y|\cos(\theta_y) + |x|\sin(\theta_x)|y|\sin(\theta_y) = |y|^2 \\
|x|\cos(\theta_x)|y|\sin(\theta_y) = |y|\cos(\theta_y)|x|\sin(\theta_x)
\end{cases}
$$

$$
\Leftrightarrow \begin{cases}
|x||y|\left[\cos(\theta_x)\cos(\theta_y) + \sin(\theta_x)\sin(\theta_y)\right] = |y|^2 \\
|x||y|\left[\cos(\theta_x)\sin(\theta_y) - \cos(\theta_y)\sin(\theta_x)\right] = 0
\end{cases}
$$

$$
\Leftrightarrow \begin{cases}
|y|\left[|x|\left(\cos(\theta_x)\cos(\theta_y) + \sin(\theta_x)\sin(\theta_y)\right) - |y|\right] = 0 \\
|x| = 0 \text{ OR } |y| = 0 \text{ OR } \left[\cos(\theta_x)\sin(\theta_y) - \cos(\theta_y)\sin(\theta_x)\right] = 0
\end{cases}
$$

$$
\Leftrightarrow \begin{cases}
|y| = 0 \text{ OR } |x|\left(\cos(\theta_x)\cos(\theta_y) + \sin(\theta_x)\sin(\theta_y)\right) = |y| \\
|x| = 0 \text{ OR } |y| = 0 \text{ OR } \cos(\theta_x)\sin(\theta_y) = \cos(\theta_y)\sin(\theta_x)
\end{cases}
$$

$$
\Leftrightarrow \begin{cases}
|y| = 0 \text{ OR } |x|\left(\frac{1}{2}\cos(\theta_x - \theta_y) + \frac{1}{2}\cos(\theta_x + \theta_y) + \frac{1}{2}\cos(\theta_x - \theta_y) - \frac{1}{2}\cos(\theta_x + \theta_y)\right) = |y| \\
|x| = 0 \text{ OR } |y| = 0 \text{ OR } \frac{1}{2}\sin(\theta_x + \theta_y) - \frac{1}{2}\sin(\theta_x - \theta_y) = \frac{1}{2}\sin(\theta_x + \theta_y) + \frac{1}{2}\sin(\theta_x - \theta_y)
\end{cases}
$$

$$
\Leftrightarrow \begin{cases}
|y| = 0 \text{ OR } |x|\cos(\theta_x - \theta_y) = |y| \\
|x| = 0 \text{ OR } |y| = 0 \text{ OR } -\sin(\theta_x - \theta_y) = \sin(\theta_x - \theta_y)
\end{cases}
$$

$$
\Leftrightarrow \begin{cases}
|y| = 0 \text{ OR } |x|\cos(\theta_x - \theta_y) = |y| \\
|x| = 0 \text{ OR } |y| = 0 \text{ OR } \theta_x - \theta_y \equiv 0 \,(\mathrm{mod}\,\pi)
\end{cases}
$$

$$
\Leftrightarrow \begin{cases}
|y| = 0 \text{ OR } |x|\cos(\theta_x - \theta_y) = |y| \\
|x| = 0 \text{ OR } |y| = 0 \text{ OR } -\sin(\theta_x - \theta_y) = \sin(\theta_x - \theta_y)
\end{cases}
$$

$$
\Leftrightarrow \begin{cases}
|y| = 0 \text{ OR } |x|\cos(k\pi) = |y|, \ k \in \mathbb{Z} \\
|x| = 0 \text{ OR } |y| = 0 \text{ OR } \theta_x = \theta_y + k\pi, \ k \in \mathbb{Z}
\end{cases}
$$

$$
\Leftrightarrow \begin{cases}
|y| = 0 \text{ OR } |x|\cos(2k'\pi) = |y| \text{ OR } |x|\cos((2k'+1)\pi) = |y| = 0 \text{ (because } \cos((2k'+1)\pi) = -1) \\
|x| = 0 \text{ OR } |y| = 0 \text{ OR } \theta_x = \theta_y + k'\pi, \ k' \in \mathbb{Z}
\end{cases}
$$

$$
\Leftrightarrow \begin{cases}
|y| = 0 \text{ OR } |x| = |y| \text{ OR } |x| = |y| = 0 \\
|x| = 0 \text{ OR } |y| = 0 \text{ OR } \theta_x = \theta_y + 2k'\pi, \ k' \in \mathbb{Z}
\end{cases}
$$

$$
\Leftrightarrow \begin{cases}
|y| = 0 \text{ OR } |x| = |y| \\
\theta_x = \theta_y + 2k'\pi, \ k' \in \mathbb{Z}.
\end{cases} \tag{11}
$$

We have just shown that $\Re(\langle \mathbf{x}|\mathbf{y}\rangle)_j = |y_j|^2$ AND $\Im(\langle \mathbf{x}|\mathbf{y}\rangle)_j = 0$ is equivalent of having $[(|y_j| = 0$ OR $|y_j| = |x_j|)$ AND $\angle x_j = \angle y_j]$. This means that the real and imaginary parts **of the inner product** between the estimate and target are respectively responsible of **the amplitude and phase matching** between the estimate and the target. Now, a solution corresponding to a null reference vector $\mathbf{y}$ could be problematic as it leads to an infinite number of choices for the estimated signal x. In fact, Choi et al. (2019) mentioned this issue and chose to work with a cosine similarity-based function in order to learn from noisy-only data. This is why it is more convenient to work with the normalized inner product loss.

## A.2 COMPLEX LAYER NORMALIZATION

Just as in complex batch normalization, complex layer normalization consists in whitening 2D vectors by left-multiplying the $\mathbf{0}$-centered data $\left(\boldsymbol{x} - \mathbb{E}[\boldsymbol{x}]\right)$ by the inverse square root of the $2 \times 2$ covariance matrix $\boldsymbol{V}$. $\tilde{\boldsymbol{x}} = (\boldsymbol{V})^{-\frac{1}{2}} \left(\boldsymbol{x} - \mathbb{E}[\boldsymbol{x}]\right)$, where the covariance matrix $\boldsymbol{V}$ is

$$\boldsymbol{V} = \left( \begin{array}{cc} V_{rr} & V_{ri} \\ V_{ir} & V_{ii} \end{array} \right)$$
$$= \left( \begin{array}{cc} \text{Cov}(\Re\{\boldsymbol{x}\}, \Re\{\boldsymbol{x}\}) & \text{Cov}(\Re\{\boldsymbol{x}\}, \Im\{\boldsymbol{x}\}) \\ \text{Cov}(\Im\{\boldsymbol{x}\}, \Re\{\boldsymbol{x}\}) & \text{Cov}(\Im\{\boldsymbol{x}\}, \Im\{\boldsymbol{x}\}) \end{array} \right).$$

Complex layer normalization is distinguished from complex batch normalization by its computation of the mean and covariance statistics over the *layer* features instead of the *batch* instances. This allows us, as in the real-valued version, to avoid estimating batch statistics during training. An intuition for batch normalization's inappropriateness is related to the sparsity, in both time and frequency domains, of speech. This is reflected in the spectrograms. Speech is temporally halting and restarting, and spectrally consists of at most a few simultaneously-held fundamentals and their discrete overtones. Mixing few speakers does not significantly change this property.

In the light of this observation, it stands to reason that statistics computed across a batch's multiple utterance mixtures are almost meaningless. Speakers within and across utterance mixtures are not controlled for volume, nor can their pauses be meaningfully aligned. Batch statistics will therefore be inappropriately driven by the mixture with the most simultaneous speakers, the loudest speaker(s), or the speaker(s) with the "dirtiest" spectrogram. Finally, in the absence of any speech, batch statistics will inappropriately boost background noise to a standardized magnitude.

The above motivates the use of exclusively intra-sample normalization techniques like Layer Normalization for speech data. Batch normalization is more appropriate for natural images, which are dense, both in space and frequency.

In addition to the fact that intra-sample normalization is more appropriate for speech signals, CLN ensures a more robust normalization of data when the number of feature maps is sufficiently large. In fact, according to the weak law of large numbers, as the sample size increases, the sample statistics approximate their expected values. Therefore, when the number of feature maps far exceeds the number of batch instances, we obtain more robust estimates because they converge, in probability, to the corresponding expected values.

## A.3 FIGURES

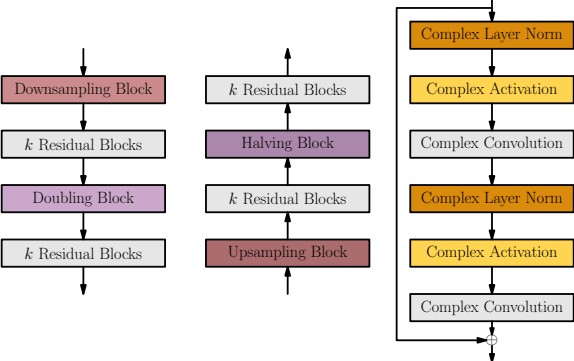

Figure 2: The basic structures of our U-Net downsampling block $D_i$ (Left) and our U-Net upsampling block $U_i$ (Middle) used respectively in the encoding and the decoding paths of Figure 1. The structure of a basic complex residual block (Right) in each of $D_i$ and $U_i$.

## A.4 Data Pre-processing and Training Details

The speech mixtures are generated using the procedure adopted in Erdogan et al. (2015); Wang et al. (2018). More precisely, the training set consists of 30 hours of two-speaker mixtures that were generated by randomly selecting sentences (uttered by different speakers) from the Wall Street Journal WSJ0 training set called `si_tr_s`. The signals are then mixed with different amplitude factors, leading signal-to-noise ratios (SNR) ranging between 0 dB and 5 dB. Using the same method, we also generated 10 hours of validation set. The test set is composed of 5 hours that were generated similarly using utterances from the different speakers belonging to the WSJ0 development set `si_dt_05`. The data sampling rate is 8KHz. Regarding the STFT parameters, a Hann window of size 256 and a hop length equal to 128 are used.

Table 2 (see section A.5) and Table 3 contain the results for the experiments conducted using the Wall Street Journal dataset. All models in Tables 2 (see section A.5) and 3 were trained using the backpropagation algorithm with Stochastic Gradient Descent with Nesterov momentum (Nesterov, 1983) set at 0.9. The gradient norm was clipped to 1. We used the learning rate schedule described in Trabelsi et al. (2017). In order to warm up the model during training, a constant learning rate of 0.01 was fixed for the first 10 epochs. From epoch 10 to 100, the learning rate was increased to 0.1. Later, an annealing of the learning rates by a factor of 10, at epochs, 120 and 150 was performed. We ended up the training at epoch 200. Models in Table 2(see section A.5 ) have been trained using a batch size of 40. Models in Table 3 have been trained using a batch size of 24 to fit in the GPU memory. All the models have been trained in parallel using 8 V100 GPUs. For all the tested losses, we used the Permutation Invariant Training criterion knows as PIT (Yu et al., 2017). The PIT criterion allows to take into account all possible assignments between the target signals and the estimated clean speeches. This is done by computing all possible permutations between the targets and the estimated clean speeches. During training, the assignment with the minimal loss is considered for backpropagation. This is due to the fact that for the synthetically mixed input, the order of the target speakers is randomly chosen and it doesn't satisfy a specific criterion. This random order in the target speakers causes the well-known label permutation problem (Hershey et al., 2015; Weng et al., 2015). The PIT criterion allows then to reduce significantly this problem by considering the output-target assignment yielding the minimal training loss. During inference, we assume that the model has learned to produce output that does not permute speeches. (Yu et al. (2017) mention that output-to-speaker assignment may change across time frames. This would have the effect of decreasing the Signal to Noise Ratio (SNR) and the Signal to Distortion Ratio (SDR) as it causes interference of speakers speeches.

## A.5 Experiments

We tried different configurations combining unitary and standard complex initializations. All of these initializations have been proposed by Trabelsi et al. (2017). It turned out that the best configuration is obtained when using a complex standard initialization for all layers, except for the convolutional layer, generating the FiLM parameters, and the first convolutional layer in the generating mask function which precedes the 2 residual blocks. For the above-mentioned convolutional layers a unitary initialization respecting the He criterion (He et al., 2015) was applied. This is not surprising as a unitary weight matrix $\in \mathbb{C}^{d \times d}$ constitutes a basis of $\mathbb{C}^d$. Therefore any complex-valued vector in $\mathbb{C}^d$, such as those representing the FiLM parameters or the masks, could be generated using a linear combination of the row vectors of that unitary matrix.

In Tables 2 and 3 we experiment with architectures that use different number of mixture transformations. Adding mixture transformations does not significantly increase the number of parameters compared to the size of the whole model. In the case where 15 transformations are adopted, the number of parameters is increased by less than 1% of the total number.

Since Table 2's first row contains baselines, they exclude our proposed masking method and loss. These baselines (both real and complex) are architecturally the same as the U-Net of Figure 1, without the FiLM, the GenerateMask and the averaging operation. A real counterpart of a complex model is one where the convolution and the normalization layers are real, the nonlinearity is plain ReLU and He init is used for the weights. The real and complex U-Nets output the masks which are complex multiplied with the mix in order to infer the clean speech of the speakers. All the complex models, whether they have approximately the same number of parameters ($\mathbb{R}$:8.45M $\approx \mathbb{C}$:7.4M),

Table 2: Speech separation experiments on two speakers using the standard setup with the Wall Street Journal corpus.

We explore different real and complex-valued model variants and report the test SDR. $k$ is the the number of residual blocks used inside the residual U-Net block (See Figure 1). Start Fmaps is the number of feature maps in the first layer of the encoding path in the U-Net. The Start Fmaps defines the width of each of the successive layers in the model. We respectively double and half the size of the layers in each of the successive downsampling and upsampling stages. The effective number of feature maps for a complex feature map is equal to the number of reported feature maps $\times 2$. This is due to the fact that it has a real and an imaginary part. The number of parameters is expressed in millions. The number of input mixture transformations is also reported. Test SDR scores for different time and spectral domain losses are inserted in the last column.

| MODEL | $k$ | START FMAPS | PARAMS | TRANSFORMS | LOSS FUNCTION | TEST SDR |
|---|---|---|---|---|---|---|
| REAL U-NET | 1 | 64 | 8.45 | 0 | $L2_{\text{freq}}$ | 4.59 |
| REAL U-NET | 2 | 64 | 14.76 | 0 | $L2_{\text{freq}}$ | 7.92 |
| COMPLEX U-NET | 1 | 32 | 4.29 | 0 | $L2_{\text{freq}}$ | 9.61 |
| COMPLEX U-NET | 2 | 32 | 7.4 | 0 | $L2_{\text{freq}}$ | **9.70** |
| COMPLEX U-NET | 2 | 40 | 11.55 | 0 | $L2_{\text{freq}}$ | **10.30** |
| COMPLEX U-NET | 2 | 40 | 11.55 | 0 | $\mathbb{C}\text{SimLoss}$ | 10.21 |
| COMPLEX U-NET | 2 | 40 | 11.55 | 0 | $L2_{\text{time}}$ | 9.31 |
| COMPLEX U-NET | 2 | 40 | 11.55 | 0 | $\cos_{\text{time}}$ | 9.34 |
| COMPLEX U-NET | 2 | 40 | 11.57 | 5 | $L2_{\text{freq}}$ | 10.58 |
| COMPLEX U-NET | 2 | 40 | 11.57 | 5 | $\mathbb{C}\text{SimLoss}$ | **10.87** |
| COMPLEX U-NET | 2 | 40 | 11.57 | 5 | $L2_{\text{time}}$ | 10.31 |
| COMPLEX U-NET | 2 | 40 | 11.57 | 5 | $\cos_{\text{time}}$ | 10.14 |
| COMPLEX U-NET | 2 | 40 | 11.61 | 10 | $L2_{\text{freq}}$ | 10.59 |
| COMPLEX U-NET | 2 | 40 | 11.61 | 10 | $\mathbb{C}\text{SimLoss}$ | **10.90** |
| COMPLEX U-NET | 2 | 40 | 11.61 | 10 | $L2_{\text{time}}$ | 10.86 |
| COMPLEX U-NET | 2 | 40 | 11.61 | 10 | $\cos_{\text{time}}$ | 10.47 |
| COMPLEX U-NET | 2 | 40 | 11.67 | 15 | $L2_{\text{freq}}$ | **10.93** |
| COMPLEX U-NET | 2 | 40 | 11.67 | 15 | $\mathbb{C}\text{SimLoss}$ | 10.91 |
| COMPLEX U-NET | 2 | 40 | 11.67 | 15 | $L2_{\text{time}}$ | 10.66 |
| COMPLEX U-NET | 2 | 40 | 11.67 | 15 | $\cos_{\text{time}}$ | 10.74 |

half ($\mathbb{R}$:8.45M; $\mathbb{C}$:4.39M) or a third, with half the depth ($\mathbb{R}$:14.76M; $\mathbb{C}$:4.39M) outperformed by a large margin their real counterparts. This shows that whether the comparison is fair, or even where advantages in terms of capacity and depth are given to the real network, it doesn't perform as well as complex models when it comes to process complex input and infer complex output. Thus, we will no longer focus on real-valued models, but, instead, will concentrate on transformations and losses that are appropriate for complex-valued models.

Three major observations can be inferred from the numbers displayed in Table 2: 1- Wider and deeper models improve the quality of separation in terms of SDRs; 2- The increase in the number of input transformations has a positive impact on the task of separating audio sources, as additional input transformations achieve higher SDR scores; 3- For a given number of input transformations, the best results are obtained with losses computed in the spectral domain. For all the experiments reported in Table 2, either the $\mathbb{C}\text{SimLoss}$ or the $L2_{\text{freq}}$ achieve the highest SDR.

The scores reported in Table 2 show that the local ensembling procedure is beneficial to the task of speech separation. This rewarding impact is confirmed in all experiments of Table 3 (See also Figure 3). As mentioned in section 6, each mask could be seen as a feature of the speaker embedding and the generated masks together constitute the whole embedding. Performing dropout on the masks might then allow to perform regularization for the retrieval and separation mechanism. Dropping out a mask is equivalent to a dropout of input mixture transformations or clean speech candidates. Since spectral loss functions yielded higher SDRs than their time-domain counterparts, we adopted them to evaluate the effect of applying different dropout rates to the input transformations. Wider and deeper models

Table 3: Experiments on two speaker speech separation using the standard setup with the Wall Street Journal corpus. We explore different numbers of input mixture transformations and different dropout rates on the latter using the training losses defined in the spectral domain. The losses in questions are $L2_{\text{freq}}$ and $\mathbb{C}$SimLoss. The number of parameters is expressed in millions. All tested models contain 44 feature maps in the first downsampling layer of the U-Net instead of 40 in Table 2. The same number of $k = 2$ residual blocks is used inside the basic structure of the residual U-Net block. SDR scores are shown in the last column.

| PARAMS | TRANSFORMS | DROPOUT | LOSS FUNCTION | TEST SDR |
|---|---|---|---|---|
| 13.97 | 0 | 0 | $L2_{\text{freq}}$ | 9.88 |
| 13.99 | 5 | 0 | $L2_{\text{freq}}$ | 10.11 |
| 14.03 | 10 | 0 | $L2_{\text{freq}}$ | 10.91 |
| 14.09 | 15 | 0 | $L2_{\text{freq}}$ | 9.92 |
| 13.97 | 0 | 0 | $\mathbb{C}$SimLoss | 9.87 |
| 13.99 | 5 | 0 | $\mathbb{C}$SimLoss | 10.64 |
| 14.03 | 10 | 0 | $\mathbb{C}$SimLoss | **11.05** |
| 14.09 | 15 | 0 | $\mathbb{C}$SimLoss | 10.82 |
| 13.99 | 5 | 0.1 | $L2_{\text{freq}}$ | 10.54 |
| 14.03 | 10 | 0.1 | $L2_{\text{freq}}$ | 10.72 |
| 14.09 | 15 | 0.1 | $L2_{\text{freq}}$ | 10.91 |
| 13.99 | 5 | 0.1 | $\mathbb{C}$SimLoss | 10.96 |
| 14.03 | 10 | 0.1 | $\mathbb{C}$SimLoss | **11.34** |
| 14.09 | 15 | 0.1 | $\mathbb{C}$SimLoss | 11.22 |
| 13.99 | 5 | 0.2 | $L2_{\text{freq}}$ | 10.67 |
| 14.03 | 10 | 0.2 | $L2_{\text{freq}}$ | 10.90 |
| 14.09 | 15 | 0.2 | $L2_{\text{freq}}$ | 10.90 |
| 13.99 | 5 | 0.2 | $\mathbb{C}$SimLoss | 11.23 |
| 14.03 | 10 | 0.2 | $\mathbb{C}$SimLoss | **11.29** |
| 14.09 | 15 | 0.2 | $\mathbb{C}$SimLoss | 11.23 |
| 13.99 | 5 | 0.3 | $L2_{\text{freq}}$ | 10.71 |
| 14.03 | 10 | 0.3 | $L2_{\text{freq}}$ | 10.06 |
| 14.09 | 15 | 0.3 | $L2_{\text{freq}}$ | 10.91 |
| 13.99 | 5 | 0.3 | $\mathbb{C}$SimLoss | **11.21** |
| 14.03 | 10 | 0.3 | $\mathbb{C}$SimLoss | 11.12 |
| 14.09 | 15 | 0.3 | $\mathbb{C}$SimLoss | 11.06 |
| 13.99 | 5 | 0.4 | $L2_{\text{freq}}$ | 10.72 |
| 14.03 | 10 | 0.4 | $L2_{\text{freq}}$ | 10.74 |
| 14.09 | 15 | 0.4 | $L2_{\text{freq}}$ | 10.83 |
| 13.99 | 5 | 0.4 | $\mathbb{C}$SimLoss | 11.09 |
| 14.03 | 10 | 0.4 | $\mathbb{C}$SimLoss | 11.08 |
| 14.09 | 15 | 0.4 | $\mathbb{C}$SimLoss | **11.12** |

with Start Fmaps = 44 and $k$=2 residual blocks are tested in the conducted experiments. Results are reported in Table 3.

In the absence of dropout and multiple transformations, we observe from the results displayed in Table 3, that wider models are not necessarily more beneficial to the separation task. The SDRs reported in the case of no mixture transformations are 9.88 and 9.87 for the wider model. These SDR scores correspond to the $L2_{\text{freq}}$ and $\mathbb{C}$SimLoss losses respectively. However, for the narrower models, SDRs of 10.30 and 10.21 were respectively reported for the same losses in Table 2. This means that wider models have the potential to overfit. On the other hand, if input transformations are taken into account, a jump in the SDR is observed. When 10 input transformations are introduced, SDR scores of 11.05 and 10.90 are recorded with the $\mathbb{C}$SimLoss and the $L2_{\text{freq}}$ losses, respectively. Lower SDR performances are recorded when ensembling is implemented with mixtures of 5 and 15 transformations, respectively. This means that the local ensembling procedure is acting as a regularizer. However, a tradeoff in terms of the number of input transformations (and so in terms of clean speech candidates) has to be made as increasing the number of input transformations might worsen the performance of the model and lead to overfitting.

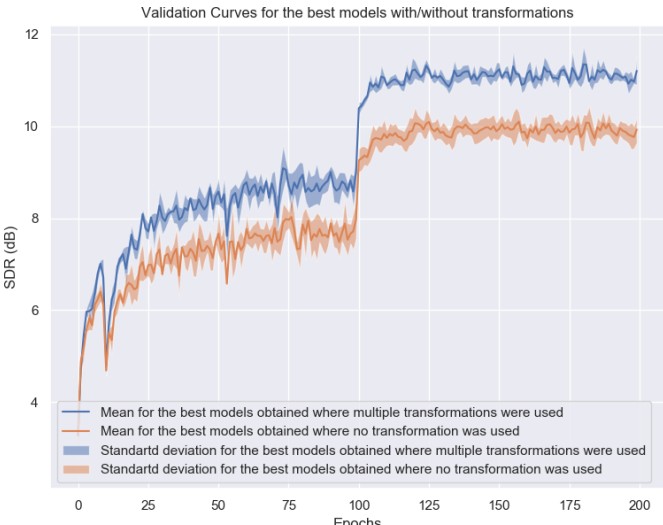

Figure 3: **Validation curves of models with and without performing multiple input transformations**. The plotted curves relate to models reported in Table 3. Models with multiple input transformations outperform those without transformations. The former achieved higher SDR scores, on average.

Dropping out the speech candidates using a small probability rate has a further regularization effect on the wider model. This could be inferred from the results reported in Table 3 (See also Figure 5). We employed different dropout rates varying from 0 to 0.4. A rate of 0.1 yielded the best result as it caused a jump of SDR score from 11.05 to 11.34. It is important to emphasize again the importance of having a compromise in terms of the number of transformations. For instance, for most of the dropout rates we experimented, a number of 10 mixture transformations yielded the highest SDRs. In all the experiments reported in Table 3, the $\mathbb{C}$SimLoss clearly outperformed the $L2_{\mathrm{freq}}$ (See Figure 4). In fact, regardless of the dropout rate and the number of input transformations employed, for wider models using the $L2_{\mathrm{freq}}$ training loss function, the SDR score did not cross the threshold of 10.91 dB. The highest SDR score obtained, when using the $L2_{\mathrm{freq}}$ loss function, is 10.93. This value corresponds to a narrower model with 15 input transformations (see Table 2).

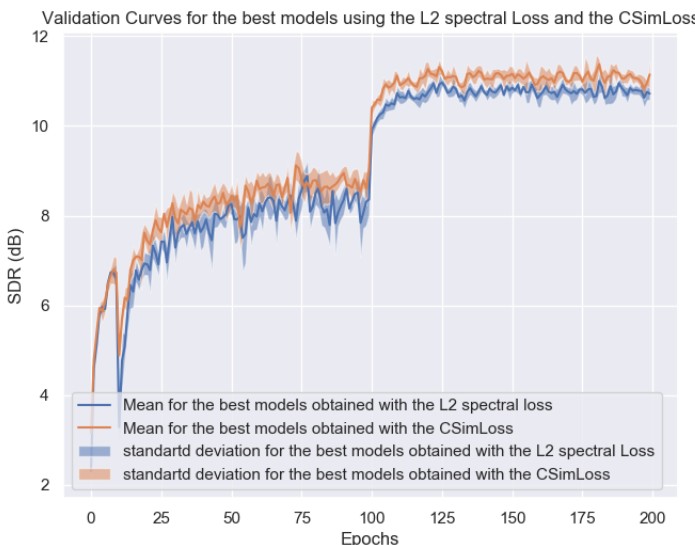

Figure 4: Validation curves of the models that yielded the highest SDRs using either the L2 spectral loss or our ℂSimLoss. The Drawn curves are related to models reported in Table 3.

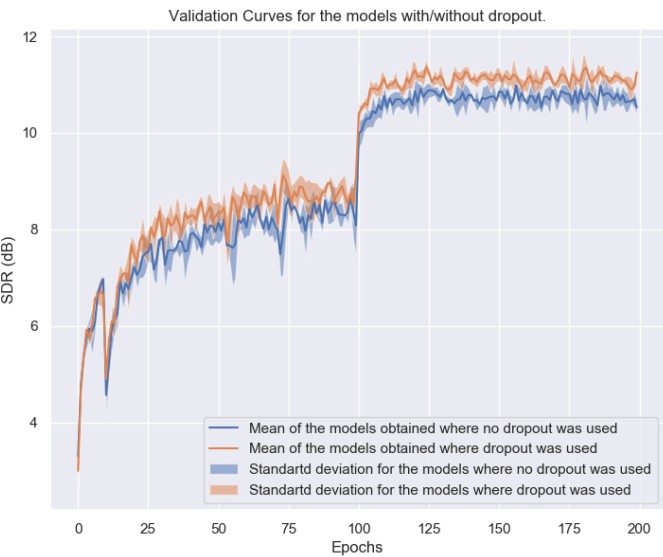

Figure 5: Validation curves of the models that yielded the highest SDRs for both cases where dropout on the input mixture transformations was used and where it was not. The Drawn curves are related to models reported in Table 3.

## A.6 State-of-the-art Table

Table 4: Listing of some existing state-of-the-art methods for two speaker speech separation

| Model | SNR | Window Size | Hop Length | Temporal Model Used | Input | SDR |
|---|---|---|---|---|---|---|
| Deep Clustering (Hershey et al., 2015) | Uniform [0, 5] dB | 32 ms = 256 samples | 8 ms = 64 samples | 2 BLSTMs | $\log|\text{STFT}(x)|$ | 6.5 |
| Deep attractor (Chen et al., 2016) | Uniform [0, 10] dB | 32 ms = 256 samples | 8 ms = 64 samples | 4 BLSTMs | $\log|\text{STFT}(x)|$ | 10.5 |
| Anchor Deep attractor (Luo et al, 2017) | Uniform [0, 5] dB | 32 ms = 256 samples | 8 ms = 64 samples | 4 BLSTMs | $\log|\text{STFT}(x)|$ | 10.8 |
| TasNet (Luo & Mesgarani, 2017) | Uniform [0, 5] dB | Time-domain segment size 5 ms = 40 samples | None | 4 BLSTMs | Raw time-domain signal | 11.1 |
| ConvTasNet (Luo & Mesgarani, 2018) | Uniform [-5, 5] dB | Time-domain conv-filter = 2 ms = 16 samples | 50% overlap 1 ms = 8 samples | Temporal Convolution Networks | Raw time-domain signal | 15.6 *(claimed)* 12.1 *(reproduced with [0,5] dB)* |
| Deep Complex U-Net (Ours, w/o extraction mechanism) | Uniform [0, 5] dB | 32 ms = 256 samples | 16 ms = 128 samples | None (No temporal recurrent model used) | $\text{STFT}(x)$ | 9.70 |
| Deep Complex U-Net (Ours, w/ extraction mechanism) | Uniform [0, 5] dB | 32 ms = 256 samples | 16 ms = 128 samples | None (No temporal recurrent model used) | $\text{STFT}(x)$ | 11.34 |

As can be seen from Table 4, state-of-the-art results in speech separation depend largely on the following:

1. The use of a model that takes into account short and long term temporal dependencies such as BLSTMs or Temporal Convolutional Networks (Bai et al., 2018). Almost all the methods since (Hershey et al, 2015) that have led to improvements in state-of-the-art speech separation have used either BLSTMs or TCN;

2. The STFT window size and hop length or the time-domain input segment size when using the raw signal. Yu et al. (2017) demonstrated that the smaller the window size, hop length are, the better the quality of separation. This probably explains Luo & Mesgarani (2017) and Luo & Mesgarani (2018) selection of very short time-domain segment sizes of 5 and 2 ms for the TasNet and ConvTasNet archtiectures respectively.

