# OpenReview forum: "Retrieving Signals in the Frequency Domain with Deep Complex Extractors"
_ICLR.cc/2020/Conference — Reject_

### Official Review · AnonReviewer1 · 2019-10-15
**Official Blind Review #1**

**Rating:** 6

**Review:**

This paper proposes a new method for source separation, by using deep learning UNets, complex-valued representations and the Fourier domain. Concretely, their contribution is : i) a complex-valued convolutional version of the Feature-Wise Linear Modulation, able to optimise the parameters needed to create multiple separated candidates for each of signal sources that are then combined using signal averaging; ii) the design of a loss that takes into account magnitude and phase while being scale and time invariant. It was then tested and compared with real-valued versions, and also some state-of-the-art methods.

Overall, I think this paper is of good quality and proposes an interesting method for this crucial task of source separation. However, I found the paper too dense and difficult to read (even if well written), and it looks like a re-submission from a journal paper of more than 8 pages. I would suggest the authors to shrink the paper so it *really* fits into the 8-pages (without important figures or important implementation details in the appendices), maybe at the cost of leaving some parts (such as old related works) out of the paper. The experiments are important here, and it is too bad that the comparison with state-of-the-art is just in the last paragraph, while the results do not seem to show any improvements compared to the other methods. The computational time might be very important here, as the claim is that FFT reduces time computation, but I did not had time to go through all the appendices.

Positive aspects:
- The work is well documented and motivated, and I found that the reflexion leading to the method is of good quality.
- Concretely, I found interesting the use of the FiLM, originally designed for another application, for minimizing the SNR of the signal sources. The motivation/proof is quite clear too.
- Equally, the motivation for the design of the new loss is clear and interesting.
- I also found important the experiments, that shows in the same table the difference between the method without the complex-valued part and with different parameter values.

Questions and remarks:
- I have to recall that I am not an expert on source separation and complex-valued deep learning. Yet, I have had difficulties in understanding the structure of the method, even if the different parts were clearly explained. The figure 1 is very useful, but I found it not clear enough and too small. I went to find some informations in the appendices, but there are too much crucial information there and I did not have time to go through all of it.
- The use of the U-net architecture is not explained (just some citations are given). What is supposed to be the output of it?
- When you say 'to be more rigorous, we can assume in the cse of speech separation that, for each speaker, there exists an impulse response such that when it is convolved with the clean speech of the speaker, it allows to reconstruct the mix' : why can we be sure that it is always possible, and why is it more rigourous?
- Why are the additive noises epsilon_i supposed to have the same E(|epsilon_i|^2|) ? Even if they are uncorrelated, what is the hypothesis behind that?
- In the CSimLoss, why (i) is the real part negative and the imaginary part positive; (ii) is the imaginary part squared?
- have you tested with a higher lambda_imag (as the larger is now the best)?
- It looks from the end of the paper that the method is still not achieving better results than the state-of-the-art. I agree with the authors as it might not be the scope of the paper, but then what is it? If it's time computation, it is not shown in the paper. If it is just a methodology, what would be required in the future to beat the best method?
- In the results, table 1, in the last 4 lines: it looks from 1st and 2nd line that the new loss CSimLoss is not very different from the L2 (9.88 compared to 9.87). The best result, in the 4th line, cannot be compared to the 3rd line as both the loss and the number of transforms are different. I then found those values in the appendices, but it would be best to show fewer parameters varying in the main paper, but show some results that can be easily compared.
- What is the importance of the first paragraph in 2.1? I was not aware of the holographic reduced representations, but I don't understand it more now, and I don't see why explaining that for 15 lines.

Small remarks:
- 'deep complex valued models have *just* started to gain momentum'... with citations beginning in 2014, I would not say 'just'.
- 'in the frequncy domain is then, ...' --> frequency + no coma
- Figure 2 is in the appendix, while in the text it is not said so. I was lost. This figure should not be in the appendix as the appendix should not have key elements, but just details that are not important for the understanding of the paper.


**Experience Assessment:**

I do not know much about this area.

**Review Assessment: Checking Correctness Of Derivations And Theory:**

I assessed the sensibility of the derivations and theory.

**Review Assessment: Checking Correctness Of Experiments:**

I carefully checked the experiments.

**Review Assessment: Thoroughness In Paper Reading:**

I read the paper thoroughly.

---

> ### Author Response · Authors · 2019-11-13
> **Response to Reviewer 1**
>
> We thank the reviewer for the useful feedback and appreciate the encouraging comments. We have considered the comments and tried to address them below.
>
> Reviewer 1: "1- I have to recall that I am not an expert on source separation and complex-valued deep learning. Yet, I have had difficulties in understanding the structure of the method, even if the different parts were clearly explained. The figure 1 is very useful, but I found it not clear enough and too small. I went to find some informations in the appendices, but there are too much crucial information there and I did not have time to go through all of it."
>
> The U-Net is not the contribution of the paper and it has been widely used in inverse problems such as image segmentation and signal reconstruction. This is why details of the residual upsampling and downsampling blocks incorporated in the U-Net are in the appendix in section A.3 and not in the main text.
>
> Sections A.4: DATA PRE-PROCESSING AND TRAINING DETAILS contains just the details about the implementation, the optimizer and the standard processing of the WSJ dataset (Hershey et al, 2015). It is not about the scientific contributions but more about the technical details and that is why it is in the appendix. We have provided our source code so the reader can have a better grasp of our method and can reproduce our work.
>
> Reviewer 1: "2- The use of the U-net architecture is not explained (just some citations are given). What is supposed to be the output of it?"
>
> In section 5 in the main text explained that the U-Net architecture we used is similar to “the complex-valued U-Net architecture used in Dedmari et al. (2018) who reported state-of-the-art results in MRI reconstruction using complex-valued raw input”. That design’s success is why we selected it as a base. The U-Net’s output is an intermediate representation that the extractor mechanism learns to use to generate clean speech candidates.
>
> Reviewer 1: "3- When you say 'to be more rigorous, we can assume in the case of speech separation that, for each speaker, there exists an impulse response such that when it is convolved with the clean speech of the speaker, it allows to reconstruct the mix' : why can we be sure that it is always possible, and why is it more rigorous?"
>
> This information is translated in equation (3) which is just the application of the equation y = s * r + epsilon, in the context of speech separation. Similar assumptions have been also been made and reported in speech-related problems such as in https://arxiv.org/pdf/1705.10874.pdf Where noisy signals are the result y(t) = s(t) ∗ r(t) + n(t). If we want to apply that assumption to speech separation, then, the analogue of the noisy constant signal y(t) becomes the mix containing all speeches, for each ith speaker that will be a clean speech s_{i}(t) an impulse response r_{i}(t) and an additive noise n_{i}(t). Now all of these assumptions correct? Of course not as George Box states: “All models are wrong but some of them are useful”. It is a wrong assumption but a useful one as it allowed us to leverage the convolution theorem and the linearity of the Fourier transform in order to deduce an extraction mechanism based on FiLM and signal averaging allowing us to increase the signal-to-noise-ratio and so, the quality of signal retrieval in the frequency domain.
>
> Reviewer 1: "4- Why are the additive noises epsilon_i supposed to have the same E(|epsilon_i|^2|) ? Even if they are uncorrelated, what is the hypothesis behind that?":
>
> We are assuming that the noise components have the same mean 0 and the same variance sigma. E[|epsilon_i|^2|] = E[epsilon_i^2] = E[(epsilon_i - 0)^{2}] = E[(epsilon_i - E[epsilon_i])^{2}] = Var[epsilon_i] = sigma. This is why the sum over i from 1 to N of E(|epsilon_i|^2|) is equal to N times E(|epsilon_i|^2|) = N sigma.
>
> Reviewer 1: "5- In the CSimLoss, why (i) is the real part negative and the imaginary part positive; (ii) is the imaginary part squared?":
>
> We have shown in section A.1 in the appendix and in section 4 that:
> a) When the real part of the normalized inner product is maximized, the match in amplitude between the estimate and the target is maximized.
> b) When the imaginary part of the normalized inner product is 0, the match in phase between the estimate and the target is maximized.
>
> Deep learning frameworks are built around minimization of some objective function. We formulate the CSimLoss so that its minimization maximizes the match between estimate and target. To do this, we minimize the negation of its real part and minimize the square of its imaginary part. The minimum of the real part’s negation is achieved at -1, and the minimum of the imaginary part’s square is achieved at $0^2 = 0$, which corresponds to our needs.

---

> > ### Author Response · Authors · 2019-11-13
> > **Response to Reviewer 1 (part 2)**
> >
> > Reviewer 1: "6- have you tested with a higher lambda_imag (as the larger is now the best)?":
> >
> > We have tested higher and lower lambdas and they have yielded instabilities during training. We agree that a hyperparameter search between these lower and bigger valued might of course yield better results.
> >
> > Reviewer 1: "7- It looks from the end of the paper that the method is still not achieving better results than the state-of-the-art. I agree with the authors as it might not be the scope of the paper, but then what is it? If it's time computation, it is not shown in the paper. If it is just a methodology, what would be required in the future to beat the best method?"
> >
> > As you can see from Table 4, which we added to the appendix, state-of-the-art results in speech separation depend largely on the following:
> >
> > a- The use of a model that takes into account short and long term temporal dependencies (such as BLSTMs or Temporal Convolutional Networks (Bai et al, 2017)). Almost all the methods since (Hershey et al, 2015) that have led to improvements in state-of-the-art speech separation have used either BLSTMs or TCN (Temporal Convolutional Networks) (Bai et al, 2018) ;
> >
> > b- The STFT window size and hop length, if using log(|STFT(x)|) or STFT(x) as input, or the time-domain input segment size, if using the raw signal. It was known (and Yu et al, 2019 conclusively demonstrated) that the smaller the window size, hop length and overlap are, the better the quality of separation. This probably explains Luo & Mesgarani (2017) (Tasnet) and Luo & Mesgarani (2018) (ConvTasnet)’s selection of very short time-domain segment sizes of 5 and 2 ms respectively.
> >
> > Since the paper seeks to demonstrate our new extraction mechanism rather than obtaining a new high score for test SDR, we have done neither of these things. We have instead used a simple U-Net and the longer window sizes and hop lengths typical of most papers in Table 4. This extraction mechanism could, however, be stacked atop a BLSTM or TCN, and is compatible with shorter window sizes and hop lengths.
> >
> > Reviewer 1: "8- In the results, table 1, in the last 4 lines: it looks from 1st and 2nd line that the new loss CSimLoss is not very different from the L2 (9.88 compared to 9.87). The best result, in the 4th line, cannot be compared to the 3rd line as both the loss and the number of transforms are different. I then found those values in the appendices, but it would be best to show fewer parameters varying in the main paper, but show some results that can be easily compared."
> >
> > We agree with the reviewer that the parameters are varying but these varying parameters are in purpose contained in Table 1 in order to show a very important conclusion. Our conclusion is that our extraction mechanism is particularly well suited for being paired with the CSimLoss objective. It can be observed from the Table that when the proposed extraction mechanism is not used (No multiple transformations and no dropout) the CSimLoss performs comparably to the L2freq loss (around 10.90db for both). However, when the extraction mechanism is introduced (FiLM with multiple transformations, signal averaging and dropout), the CSimLoss significantly outperforms the L2freq loss and yields the best results. L2Freq is unable to yield better performance when the extraction mechanism is introduced (10.93 dB; unchanged) whereas CSimLoss does (11.34 dB).

---

> > > ### Author Response · Authors · 2019-11-13
> > > **Response to Reviewer 3 (part 3)**
> > >
> > > Reviewer 1: "9- What is the importance of the first paragraph in 2.1? I was not aware of the holographic reduced representations, but I don't understand it more now, and I don't see why explaining that for 15 lines."
> > >
> > > HRR is intimately related to our extraction mechanism. As we stated in section 2.1, HRRs are a form of key-value storage. Information is stored in the memory by entangling the key and the value together with circular convolution and then summing this bound representation into a storage vector. To retrieve a value by its associated key, it is enough to perform the same circular convolution with the inverse of that key. The result is, in expectation, the value plus some Gaussian noise (so the retrieved representation quite noisy). A different version of this was implemented in the associative LSTM paper, Danihelka et al. (2016), in order to augment the memory of the RNN. According to the convolution theorem, performing element-wise multiplication in the frequency domain is dual to performing the circular convolution in the time domain. Element-wise multiplication between the inverse of the key and the storage vector would allow to retrieve the representation in question only when both representations are encoder or converted to the frequency domain. Danihelka et al. (2016)  however, have not relied on FFTs in order to convert the temporal signals to the frequency domain.  They have assumed that element-wise multiplication is by itself is enough in order to retrieve the desired representation. The authors have also relied on summing random permutations of the bound representations by creating randomly permuted copies of the storage vector in order to decorrelated the noises during the retrieval. This is akin to our retrieval mechanism in the frequency domain as it is based on element-wise multiplication in the frequency domain and on decorrelating noises so that their mean is 0.

---

### Official Review · AnonReviewer3 · 2019-10-23
**Official Blind Review #3**

**Rating:** 3

**Review:**

This work researches the deep complex-valued neural networks. Specifically, it proposes a new signal extraction mechanism that operates in frequency domain and applies to address the speech separation issue. Also, a function is proposed to explicitly consider both the magnitude and phase information of a signal. Related work on learning representation in frequency domain and speech separation is well introduced. Theoretical analysis is conducted to show the motivation and connection to signal processing. The architecture of the deep neural networks is presented in details, with the elaboration of the complex mask generation. Experimental study is conducted on a benchmark dataset to compare the proposed complex networks with those using real-part values only to demonstrate the improvement.

The rating is 3: Weak Reject considering that the novelty is limited and the experimental study is weak.

1. The significance of the theoretical analysis in Eq.(1) to Eq.(4) needs to be better explained. Currently, they seem to be some straightforward results in the field of signal processing;
2. The proposed CSimLoss is interesting. However, its effectiveness seems to be limited as demonstrated in Table 1. It can be found that the CSimLoss in some cases is only comparable (or even inferior) to the L2freq loss;
3. The mask generation proposed in Section 6 conceptually is largely an attention mechanism that has been widely applied in deep networks;
4. The experimental comparison in Table 1 is limited, although some improvements have been demonstrated. This work shall also make a comparison with some of the existing methods on speech separation as described in Section 2.2.
5. It was mentioned in the last paragraph of Section 7 that (Shi et al. 2019) uses different data preparation than this paper. Can this paper use the same data preparation as (Shi et al. 2019) and perform some comparisons?
6. Why did (Shi et al. 2019) achieve better SDR (12.1 vs. 11.3) than the proposed method using standard setup?
7. What if the mechanism of mask generation is also applied to Real U-Net? How much improvement can this bring?


**Experience Assessment:**

I do not know much about this area.

**Review Assessment: Checking Correctness Of Derivations And Theory:**

I assessed the sensibility of the derivations and theory.

**Review Assessment: Checking Correctness Of Experiments:**

I assessed the sensibility of the experiments.

**Review Assessment: Thoroughness In Paper Reading:**

I read the paper at least twice and used my best judgement in assessing the paper.

---

> ### Author Response · Authors · 2019-11-13
> **Response to Reviewer 3 (Part 1)**
>
> We thank the reviewer for the useful feedback. We will attempt here to highlight those aspects of the paper we believe are novel, and defend our experimental study’s setup.
>
> Reviewer 3: "1. The significance of the theoretical analysis in Eq.(1) to Eq.(4) needs to be better explained. Currently, they seem to be some straightforward results in the field of signal processing;
> Answer by the authors":
>
> Section 3 and Equations (1) to (4) are related to signal processing, but serve to justify the use of FiLM and signal averaging by demonstrating how a combination of both increases the Signal-To-Noise ratio (SNR) of retrieved signals and the quality of retrieval. By elaborating such a demonstration, we make a bridge between the proposed extraction approach, that combines FiLM and signal averaging, and the literature in signal processing. Such a connection allows us to show that the proposed approach is a principled method for signal extraction in the frequency domain. More precisely, in section 3, we explain how:
> 1- Using Film makes sense in the context of signal retrieval in the frequency domain as it learns a scaling (Γ) and a shifting representation (B) of a noisy signal F(y) in order to retrieve the clean signal F(s);
> 2- Approximating F(s) by performing the operation of signal averaging allows one to increase the signal to noise ratio of the retrieved signal.
>
> We provide the following explanation to help understand the method:
>
> Equation (1) gives the expression of the Fourier transform of a noisy signal y. By leveraging the linearity of the Fourier Transform and Convolution Theorem we get equation (1). Equation (2) is just the expression of the clean signal in terms of the Fourier transforms of the noisy signal, the impulse response and the noise component. In equation (2), [1 / F(r)] and -[F(epsilon) / F(r)] are  respectively scaling and shifting representations of F(y). Now, Film (Perez et al., 2017) is a mechanism that allows to infer a scaling (Γ) and a shifting representation (B) given an input representation. This is why we use FiLM as it allows to infer Γ = [1 / F(r)] and B = -[F(epsilon) / F(r)] given the noisy mix and the output of the U-Net. Equations (1) and (2) allow us to express F(s) in terms of  Γ and B. Equation (3) is just the application of equation (2) in the context of speech separation where multiple speakers are involved in the mix and we have to retrieve the clean speech of each of the distinct speakers. We assume then, that, for each speaker, there exists an impulse response r_{i} and an additive noise epsilon_{i} such that they allow to reconstruct the original constant mix. This leads to the expression of Equation (3).
>
> Now, indeed the text between Equation (3) and Equation (4) is related to a very well-known result in signal processing which explains how signal averaging increases the signal to noise ratio of a given signal by a factor of N. This text provides to readers unfamiliar with signal averaging and signal processing a clear motivation for using signal averaging in the context of signal retrieval in the frequency domain. Equation (4) then gives the expression of F(s) when such an operation is implemented.
>
> Reviewer 3: "2. The proposed CSimLoss is interesting. However, its effectiveness seems to be limited as demonstrated in Table 1. It can be found that the CSimLoss in some cases is only comparable (or even inferior) to the L2freq loss; You have has also mentioned in your 4th remark : 4. The experimental comparison in Table 1 is limited, although some improvements have been demonstrated.":
>
> Table 1 provides a synopsis of the most important results obtained for the task of speech separation. A more exhaustive set of experiments are provided in the appendix. As mentioned in the beginning of section 7, the complete results and the extended empirical analysis can be found in the appendix in section A.5 where the list of all the experiments is contained in big tables (Tables 2 and 3).
>
> Also, as mentioned in the analysis in section 7, and as it can be observed from Tables 2 and 3, our extraction mechanism is particularly well suited for being paired with the CSimLoss objective. We draw this conclusion because when the proposed extraction mechanism is not used the CSimLoss performs comparably to the L2freq loss. However, when the extraction mechanism is introduced (FiLm with multiple transformation, signal averaging and dropout), the CSimLoss significantly outperforms the L2freq loss and yields the best results.

---

> > ### Author Response · Authors · 2019-11-13
> > **Response to Reviewer 3 (part 2)**
> >
> > This is explained in detail at the end of section A.5 in the appendix where we state that we can also observe from Table 3 that, for all wider models and when multiple transformations and dropout rates were introduced, the CSimLoss consistently outperformed the L2Freq loss (It could also be observed in Figure 4 that the best CSimLoss models outperforms the best L2Freq models). More precisely, when wider models (with wider feature maps) are introduced in Table 3, when using the L2Freq loss, the SDR score did not cross the threshold of 10.91 dB while it reached 10.93 dB for narrower models (the ones reported in Table 2 that have less feature maps and so less parameters). When it comes to the CSimLoss, while it performed almost as well as L2Freq for narrower models, as it reached 10.91 dB in Table 2, jumps in SDR were recorded for the CSimLoss in Table 3 as it reached 11.05 dB when no dropout was introduced and then a jump to 11.35 db in terms of SDR was recorded when a dropout rate of 0.1 was used. This demonstrates that:
> > a- The CSimLoss is implementing inherently a regularization mechanism allowing to avoid overfitting when wider models are used;
> > b- The CSimLoss is operating better when our extraction mechanism (multiple transformations and dropout rate) is introduced. And there, the best SDR of 11.35 db is reported. This is why, as we stated in section 7, the CSimLoss objective is therefore particularly well suited for being paired with the CSimLoss objective.
> >
> > Reviewer 3: "3. The mask generation proposed in Section 6 conceptually is largely an attention mechanism that has been widely applied in deep networks;":
> >
> > The attention mechanism, FiLM and the masking method are three distinct decoding methods (or conditional mechanisms) that operate differently to achieve a common goal which is inferring an appropriate representation for the task at hand. They constitute more than just a simple attention mechanism. More precisely, attention is a conditional mechanism that, given an input representation, learns a categorical distribution in order to compute a weighted average of another encoded representation. Our approach does not implement an attention mechanism as it doesn’t compute a categorical distribution in order compute a weighted average. Masking is a conditional mechanism that, given an input representation, allows one to learn a mask (an element-wise multiplication filter) that filters another representation. FiLM is a conditional mechanism that learns an affine transformation of a representation given an input representation. In our case, (as described in Figure 1 and Algorithm 1) FiLM is used in order to conditionally generate affine transformations of the input mix. Importantly, these affine transformations are used to conditionally generate complex masks. These masks are then used to filter the mix input in order to retrieve the clean speech of the distinct speakers. As multiple affine transformations are inferred using FiLM, this allows us to obtain different candidates for the estimate of the clean speech for a given speaker. Furthermore, we use dropout on these multiple affine transformations during training to encourage the different candidates to become decorrelated. The different candidates are then averaged to obtain an approximation of the clean signal. This is why our method is reminiscent of ensemble methods. As one can see from our summaries above, there are many conditional processing techniques that have been used in deep neural networks for decoding purposes. To summarize: our method uses many different steps to conditionally process internal representations using local ensembles of affinely transformed complex masking operations. This is why our approach is fundamentally different from an attention mechanism.
> >
> > Reviewer 3: "4-a. The experimental comparison in Table 1 is limited, although some improvements have been demonstrated."
> >
> > We have already answered that in 2 (see our answer in 2- where we mentioned that table 1 contains a summary of the most important results and that The complete results and the extended empirical analysis can be found in the appendix in section A.5 where the list of all the experiments is contained in big tables (Tables 2 and 3)).

---

> > > ### Author Response · Authors · 2019-11-13
> > > **Response to Reviewer 3 (part 3)**
> > >
> > > Reviewer 3: "4-b. This work shall also make a comparison with some of the existing methods on speech separation as described in Section 2.2.":
> > > Please take a look at our revised manuscript where we have added a Table 4 in section A.6 in the appendix listing some of the existing methods on speech separation and their corresponding performance. We would like to draw your attention to the fact that all of the methods listed in Table 4 are fundamentally different from our approach in important ways:
> > > a-They do not operate in the frequency domain as they discard the phase information and operate in the log scale of the magnitudes of the fourier transform or in the time domain as it is the case for Tasnet and ConvTasnet.
> > > b- All of the models listed in Table 4 take into account short and long term temporal dependencies by adding BLSTMs to their models or by using TCN (Temporal Convolutional Networks (Bai et al, 2018)) such is the case of ConvTasnet. In our case, we use a U-Net which does not take into account long-term temporal dependencies. We are not interested in performance gain induced by external models such as temporal ones, but rather by the improvement caused by our extraction framework (extraction mechanism + loss) that could be incorporated into any type of model.
> > >
> > > Reviewer 3: "5. It was mentioned in the last paragraph of Section 7 that (Shi et al. 2019) uses different data preparation than this paper. Can this paper use the same data preparation as (Shi et al. 2019) and perform some comparisons?":
> > >
> > > (Shi et al, 2019) is based on (Luo & Masgarani, 2018). Both chose a specific, non-standard preparation of the data, specifically a different SNR range than the one in the canonical reference (Hershey et al, 2015) which can be found here (http://www.merl.com/demos/deep-clustering). Their preparation has the effect of raising headline SDR scores, making it unfair to compare them to all other works. To permit apple-to-apple comparisons we have reproduced (Luo & Masgarani, 2018) with the data preparation that has gained a consensus in the community. The SDR score posted by (Luo & Masgarani, 2018) predictably fell significantly. A similar drop is to be expected for (Shi et al, 2019).
> > >
> > > Reviewer 3: "6. Why did (Shi et al. 2019) achieve better SDR (12.1 vs. 11.3) than the proposed method using standard setup?":
> > >
> > > As can be seen from Table 4, which we added to the appendix, state-of-the-art results in speech separation depend largely on the following:
> > >
> > > a- The use of a model that takes into account short and long term temporal dependencies (such as BLSTMs or Temporal Convolutional Networks (Bai et al, 2017)). Almost all the methods since (Hershey et al, 2015) that have led to improvements in state-of-the-art speech separation have used either BLSTMs or TCN (Temporal Convolutional Networks) (Bai et al, 2018) ;
> > >
> > > b- The STFT window size and hop length, if using log(|STFT(x)|) or STFT(x) as input, or the time-domain input segment size, if using the raw signal. It was known (and Yu et al, 2019 conclusively demonstrated) that the smaller the window size, hop length and overlap are, the better the quality of separation. This probably explains Luo & Mesgarani (2017) (Tasnet) and Luo & Mesgarani (2018) (ConvTasnet)’s selection of very short time-domain segment sizes of 5 and 2 ms respectively.
> > >
> > > Since the paper seeks to demonstrate our new extraction mechanism rather than obtaining a new high score for test SDR, we have done neither of these things. We have instead used a simple U-Net and the longer window sizes and hop lengths typical of most papers in Table 4. This extraction mechanism could, however, be stacked atop a BLSTM or TCN, and is compatible with shorter window sizes and hop lengths.
> > >
> > > Reviewer 3: "7. What if the mechanism of mask generation is also applied to Real U-Net? How much improvement can this bring?":
> > >
> > > As can be observed in Table 1, the complex U-Net outperforms by a very significant margin the real-valued U-Net. Neither of these baselines incorporate our extraction mechanism. The proposed extraction mechanism operates on complex-valued representation. The output of the U-Net should therefore be complex so that the extraction mechanism can process it. (Trabelsi et al, 2017) have shown that hybrid models (ones combining both real and complex representation) do not perform as well as models that either operate fully on complex representations or fully on real-valued representations. Given the performance of complex-valued U-Nets compared to the real-valued ones and given that hybrid models do not perform as well as fully-real/-complex models, it is unlikely that a hybrid model combining a real U-Net and our extraction mechanism would achieve performance comparable to our fully-complex pipeline.

---

> > > > ### Public Comment · ~Manuel_Pariente1 · 2020-03-13
> > > > **The data used in (Luo & Mesgarani, 2018) and (Shi et al, 2019) is standard**
> > > >
> > > > I'd like to point out that the "accusation" that both  (Luo & Mesgarani, 2018) and (Shi et al, 2019) use non-standard data preparation is wrong.
> > > >
> > > > If noise-free 2-speaker mixtures are considered, "SNRs between 0dB and 5dB" is strictly equivalent to "SNRs between -5dB and 5dB".
> > > > If speaker S1 has SNR of +XdB, S2 will have -XdB, that's in the definition.
> > > >
> > > > If we go look at the mix_2_spk_tr.txt file that can be downloaded from the official data prepatation recipe (https://www.merl.com/demos/deep-clustering/create-speaker-mixtures.zip), the mixing weight associated to s1 is always positive. Effectively, s1 is always mixed between 0dB and 5dB, hence s2 is always mixed between 0dB and -5dB
> > > >
> > > > Many implementation were able to reproduce the results of ConvTasnet, with official data preparation recipes. And the authors of the original paper also used the official mixing scripts.
> > > >
> > > > Accusing other papers to cheat on their data preparation to increase their performances is pretty strong. Please be sure that you are right about it next time.
> > > > Please correct the paper accordingly.
> > > >
> > > > Best,
> > > > Manuel Pariente

---

### Official Review · AnonReviewer2 · 2019-10-24
**Official Blind Review #2**

**Rating:** 6

**Review:**

The authors propose complex valued neural networks to perform audio source separation in the Fourier domain. The adapt a well known U-Net architecture to the task by introducing a complex-valued FiLM layer and a new complex similarity loss that explicitly takes magnitude and phase into account. They motivate the use of complex values well and demonstrate performance and parameter efficiency improvements over real-valued baselines. Importantly, they do not need to perform spetrogram inversion because their network works natively in the complex domain. Despite the quantitative improvements over spectral models, they still slightly underperform the ConvTasNet baseline that operates directly in the waveform domain (which was slightly misleading to not include in the table). The authors perform an extensive hyperparameter search to tune the model and provide sufficient detail to reproduce their experiments. While the results did not improve upon the best baseline, they do provide further evidence to the value of using complex-valued neural networks to handle complex-valued data (where phase and synchronicity matter), which I believe will be of value to the ICLR community, and thus I lean slightly in favor of acceptance.

**Experience Assessment:**

I have read many papers in this area.

**Review Assessment: Checking Correctness Of Derivations And Theory:**

I assessed the sensibility of the derivations and theory.

**Review Assessment: Checking Correctness Of Experiments:**

I assessed the sensibility of the experiments.

**Review Assessment: Thoroughness In Paper Reading:**

I read the paper at least twice and used my best judgement in assessing the paper.

---

> ### Author Response · Authors · 2019-11-13
> **Response to Reviewer 2**
>
> We thank the reviewer for the useful feedback and appreciate the encouraging comments.
>
> To address quickly existing state of the art methods, we have added a Table 4 to our paper, summarizing models in the literature and including ConvTasNet.
>
> One important clarification that Table 4 provides is that the various methods use different data preparations and parameters. In particular, ConvTasNet stands out for its use of a non-standard preprocessing of the data, unlike that of the other methods and likely to improve its headline SDR score. Our ConvTasNet reproduction with the standard data preprocessing protocol yielded a significantly lower score.
>
> ConvTasNet also stands out for its use of smaller windows and hop lengths, which favour it relative to other papers (Yu et al, 2017). We have used window sizes and hop lengths more typical of the other papers, facilitating comparisons.

---

### Decision · Program_Chairs · 2019-12-19

**Decision:**

Reject

**Comment:**

The paper discusses audio source separation with complex NNs.  The approach is good and may increase an area of research.  But the experimental section is very weak and needs to be improved to merit publication.